# A flexible optoacoustic blood 'stethoscope' for noninvasive multiparametric cardiovascular monitoring

Haoran Jin[1,2,6], Zesheng Zheng [1,3,6], Zequn Cui [4], Ying Jiang [4], Geng Chen [4], Wenlong Li [4], Zhimin Wang[5], Jilei Wang[4], Chuanshi Yang[1], Weitao Song[1], Xiaodong Chen [4] ✉ & Yuanjin Zheng[1] ✉

Quantitative and multiparametric blood analysis is of great clinical importance in cardiovascular disease diagnosis. Although there are various methods to extract blood information, they often require invasive procedures, lack continuity, involve bulky instruments, or have complicated testing procedures. Flexible sensors can realize on-skin assessment of several vital signals, but generally exhibit limited function to monitor blood characteristics. Here, we report a flexible optoacoustic blood 'stethoscope' for noninvasive, multiparametric, and continuous cardiovascular monitoring, without requiring complicated procedures. The optoacoustic blood 'stethoscope' features the light delivery elements to illuminate blood and the piezoelectric acoustic elements to capture light-induced acoustic waves. We show that the optoacoustic blood 'stethoscope' can adhere to the skin for continuous and noninvasive in-situ monitoring of multiple cardiovascular biomarkers, including hypoxia, intravascular exogenous agent concentration decay, and hemodynamics, which can be further visualized with a tailored 3D algorithm. Demonstrations on both in-vivo animal trials and human subjects highlight the optoacoustic blood 'stethoscope"s potential for cardiovascular disease diagnosis and prediction.

Multifunctional personal healthcare with continuous monitoring is the key to advancing the next generation of precision medicine. Currently, by collecting a broad spectrum of physical or chemical information from superficial skin, an emerging class of flexible sensors showcases their abilities for in situ analysis of body temperature[1], surface electrophysiological signals[2], sweat[3,4], saliva[4], etc. Continuous and noninvasive monitoring of clinically significant physiological signatures from deeply embedded blood vessels, however, remains a challenge. Blood carries abundant clinical information but is challenging to

access noninvasively due to a skin barrier. It is of great significance to directly measure multiple physical and chemical blood dynamics properties for the early detection or treatment of cardiovascular diseases[5,6]. For example, continuous blood oxygen saturation assessment, especially on patients, elders, and infants, can help to reflect the patient's oxygen supply capacity[7]; the tracking of chemical compositions, e.g. exogenous agent concentration in blood, offers meaningful pharmacokinetic information[8]; furthermore, the hemodynamics analysis can be harnessed for cardiovascular disease diagnosis[9].

[1]School of Electrical and Electronic Engineering, Nanyang Technological University, Singapore 639798, Singapore. [2]The State Key Laboratory of Fluid Power and Mechatronic Systems, School of Mechanical Engineering, Zhejiang University, Hangzhou 310027, China. [3]Institute of Microelectronics, Agency for Science, Technology and Research, Singapore 138634, Singapore. [4]School of Materials Science and Engineering, Nanyang Technological University, Singapore 639798, Singapore. [5]School of Physical and Mathematical Sciences, Nanyang Technological University, Singapore 637371, Singapore. [6]These authors contributed equally: Haoran Jin, Zesheng Zheng. ✉e-mail: chenxd@ntu.edu.sg; yjzheng@ntu.edu.sg

The noninvasive monitoring of blood physiochemical information is currently achieved using optical or acoustic sensors. With the advances in optoelectronics, optical sensors that receive light reflections from blood can be crafted into flexible and thin films, thereby allowing close adhesion to the skin[10,11]. Therefore, cardiovascular-related parameters, including oxygen saturation[12], glucose level[13], blood pressure[14], and blood flow[15] can be optically monitored. However, due to the rapid photon diffusion and scattering inside human tissues[16], optical sensors can only statistically read out the light radiation region's gross physiological information but cannot resolve any specific blood vessel in deep tissue. Similarly, the detection of the blood flow, based on thermal transportation properties[17,18], for skin physiology investigation only enables access to the superficial vasculature. Besides the optical sensors, flexible vibration sensors that detect ultrasound pulses transmitted from the blood vessels also can evaluate cardiovascular-related hemodynamics information, which has been well miniaturized and made skin-conformable[19,20]; Deep vascular measurement at a single blood vessel resolution can be realized using vibration sensors[21–23]. Nevertheless, vibration measurements involving ultrasound techniques are only sensitive to the mechanical properties of blood and unable to perceive the variation in blood physiological biomarkers (Supplementary Note 1). Other noninvasive methods like the electrochemical method[24] can infer blood physiological parameters indirectly from bodily fluids like sweat, tears, and saliva. Still, the composition of external fluids is only partially dependent on the blood[25] and also cannot reflect in-situ blood status

and blood vessel morphology instantly (more comparisons in Supplementary Note 2). In short, current blood sensors are unable to locate the blood vessels underneath the skin to provide direct information on distributed blood properties. Beyond that, they can only offer a single physiological vital measurement, capping their capability toward comprehensive, multi-dimensional assessment of cardiovascular health status.

Optoacoustic imaging is a hybrid imaging modality combining the advantages of deep penetration of acoustics and rich spectral contrast of optics, which can sense different blood parameters in deep tissues[26–30] but still requires a bulky and rigid sensor head and testing front-end. Here, we present a flexible optoacoustic blood 'stethoscope' (OBS), featuring noninvasive, and continuous 3D multiparametric cardiovascular monitoring capabilities (Fig. 1a). Benefiting from the attachable and continuously working properties, the OBS is able to dynamically and functionally analysis of intrinsic blood and vascular more than structural image and display in conventional OA systems. It can detect multiple physiological vital signs such as hypoxia, exogenous agent decay, vascular compliance, endothelial dysfunction, and reconstruct a 3D blood vessel model with our tailored algorithm, providing remarkable insight into cardiovascular functions from various aspects. The OBS is rooted in the optoacoustic (OA) effect[27]: it delivers light into blood, triggering mechanical vibration of blood chromophore, and receives the blood signal in the form of acoustic waves. The OBS comprises a micro-lens array with pre-drilled holes for light delivery and an alternant piezoelectric sensor array as the

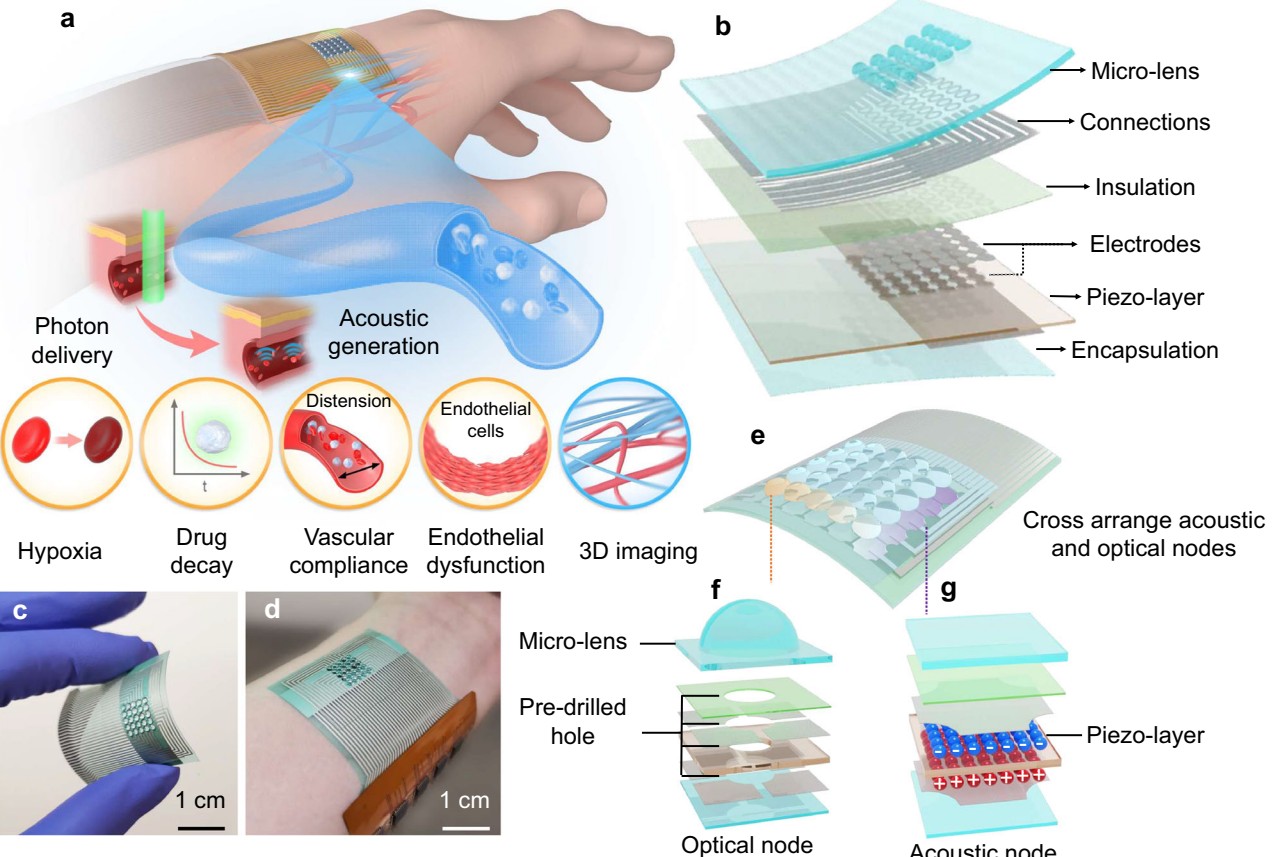

**Fig. 1 | Design and working principle of the OBS. a** Schematic illustration of the OBS laminated on the surface of the hand. The OBS is based on the optoacoustic effect and can monitor hypoxia, exogenous agent decay, vascular compliance, and endothelial dysfunction. Meanwhile, OBS can map the blood information to a 3D distribution model. **b** Schematic layout of the OBS: a multi-layered integration of optical (micro-lens), and acoustic (piezo-layer) elements with connection,

insulation, electrode, and encapsulation layers. See Methods and Supplementary Figs. 1–2 for the detailed OBS fabrication process. **c** Photograph of the OBS under bending. **d** Photograph of the OBS with flexible connecting circuits closely attached to the human wrist. **e** The cross-distributed acoustic and optical nodes. **f, g** An exploded view of a single optical (**f**) and acoustic (**g**) node.

acoustic receiver. We have chosen to term our device a blood 'stethoscope' because, akin to a traditional stethoscope, it 'listens' to the acoustic waves to characterize blood properties. This terminology is used to represent the information acquisition process and its outputs, similar to a conventional stethoscope, while it also signifies the device's unique capability to capture imaging data that serves primarily to acquire these vital signs. The flexible OBS is designed to increase patient comfort during measurement, which is a fundamental advancement over other existing OA implementations. In-vivo animal experiments and on-body human trials demonstrate that the OBS could noninvasively locate the blood vessels and continuously monitor rich blood signatures directly correlated to multiple cardiovascular conditions, such as hemodynamics, hypoxia, and intravascular exogenous agent metabolism.

Here, the OBS provides an unprecedented solution to measure venous distensibility and arterial endothelial function for cardiovascular study. Meanwhile, it can distinguish veins and arteries in the deep tissue and record their corresponding hypoxic states, which is infeasible using traditional optical and ultrasound solutions. Our proposed flexible scheme avoids conventional complex fixture setup and is more versatile and accurate for cardiovascular disease prediction and diagnosis.

## Results

The OBS device is made lightweight and flexible. It integrates multi-layered optical delivery and acoustic receiving elements with dense interconnections and array electrodes (see Methods and Supplementary Figs. 1, 2 for detailed fabrication processes). Our OBS comprises 25 optical and 36 acoustic nodes for precisely positioning a single blood vessel, with its stack structure exploded in Fig. 1b. Poly-dimethylsiloxane (PDMS)-fabricated convex micro-lens array works as the optical focusing component to deliver light, and a piezoelectric polyvinylidene fluoride (PVDF) film is implemented as the acoustic receiver (See Supplementary Fig. 3 for dimensional details). All the layers are soft and bendable under external strain (Fig. 1c), and the OBS can cling to the hand, wrist, or any other part of the body (Fig. 1d). In the optical and acoustic node design, each optical node is surrounded by four acoustic nodes, and each acoustic node has four light focal spots around its corners. This cross-distributed design of the nodes enables efficient light-acoustic coupling (Fig. 1e). The light illuminating the OBS will be focused into small spots with the optical node (Fig. 1f) and pass through the pre-drilled holes in the laminated films, then it penetrates human skin and stimulates the ultrasonic emission from the blood, and finally, the light-induced ultrasound waves are captured by the acoustic receiving nodes (Fig. 1g). Bespoke fiber solutions are employed for light delivery without compromising the flexibility of OBS to accommodate the skin bending for continuous daily monitoring (Supplementary Note 3). The OBS is versatile to monitor and image various blood characteristics by targeting different blood vessels or selecting different light wavelengths.

PVDF is selected as the piezoelectric layer in the acoustic sensor for two reasons. Firstly, the material is soft by itself, while other bulk piezoelectric materials borrowing flexibility from the soft substrates, named island-bridge structure, are complex to fabricate[21]. Most island-bridge structures require wide inter-node spacing for the bending-supported soft substrates. Therefore, the receiving sensors are sparsely distributed, leading to low efficiency in the area utilization both acoustic-wisely and optical-wisely. Secondly, PVDF is excellent in ultrasound receiving due to its high piezo-stress constant and low acoustic impedance. In traditional ultrasound applications, PVDF is relatively weak in ultrasound emission compared with other bulk piezoelectric materials. Still, for our applications, unlike the ultrasound actuator, the ultrasound waves are excited by laser illumination, and PVDF is only responsible for acoustic receiving. Hence, PVDF is highly suited for the OBS's design therein. The metal electrodes patterned on

the PVDF are made from silver, serving as a main conductive layer that provides excellent electrical characteristics while reflecting light out of the aperture. Thus, it could reduce the undesired thermal damage from redundant illumination on electrodes and other polymer layers. PDMS is regarded as the most appropriate material for the micro-lens array due to its easy-to-mold capability and high transparency in a broad spectrum range (400–1100 nm, transmittance >90%)[31].

## Optical and acoustic elements design and characterization

The OBS requires an optical element for light delivery and an acoustic element for detecting stimulated ultrasound waves from blood. The optimal functioning of the OBS requires a dedicated design of the optical and acoustic elements. Unfortunately, the tradeoff in the area between these two elements should be carefully balanced for the best detecting efficiency: Increasing the acoustic receiving areas may block the light and reduce the amount to be delivered into tissue, while overgrowing the coverage of optical elements inevitably degrades the acoustic sensitivity and may lead to reconstruction artifacts. Traditionally, optoelectronic devices adopt transparent substrates and electrodes (ITO, indium tin oxide) for optimal light delivery. However, such a solution may severely deteriorate the bendability of OBS and the acoustic receiving performances due to its high resistance. Therefore, we propose an extraordinary micro-lens array with nested pre-drilled holes for light delivery in our OBS. The micro-lens array, featuring 5 × 5 hemispherical lenses made from PDMS, can focus incoming light and deliver it through pre-drilled holes into tissue. The thickness of the micro-lens array layer should be carefully determined, which affects the light delivery efficiency (Supplementary Note 4) and the acoustic characteristics and device flexibility. A thin layer cannot guarantee sufficient light delivery, while an excessively thick one will destroy the flexibility of the OBS. Benefiting from the high transparency of PDMS, the total optical transmission coefficient of the OBS could reach up to >65% with a PDMS thickness of 500 μm, which is comparable to other piezoelectric devices with transparent electrodes (Supplementary Note 5). Our optical element design can maintain excellent light transmission without deteriorating the piezoelectric sensor's acoustic sensitivity. For other transparent piezoelectric devices, increasing the piezoelectric layer thickness can enhance acoustic sensitivity but at the cost of reduced light transmission. Our design allows light delivery through a thicker piezoelectric layer via pre-drilled holes to improve acoustic sensitivity without affecting light transmission. Similarly, the transparency of ITO electrodes is also determined by the electrode thicknesses. For 400 nm ITO, its transparency is close to our light delivery solution, while the electric resistance is up to several orders of magnitude higher than Ag. In our solutions, the Ag electrodes are 7 μm in thickness, and the interconnection is made of 10 μm silver paste to keep a low resistance in long-distance transmission. According to refs. 32,33, a considerable resistance not only induces a signal loss in circuit transmission but also constructs an output RC low-pass filter, as a side effect that causes acoustic resolution deterioration.

For OBS, achieving uniformity of the delivered light beam in a very short distance is critical to avoid any additional pre-diffusion layer and thus maintain minimum thinness of the entire device. To understand better the superiority of our optical element design in the aspect of light delivery, the Monte Carlo simulation was conducted to compare light fluence distributions of our micro-lens array method (Fig. 2a) with other mostly adopted schemes, including the side illumination method (Fig. 2b)[34,35] and the center window method (Fig. 2c)[36]. The light transmission medium is assumed to be dermis tissue with optical scattering and absorption coefficients of 357 cm$^{-1}$ and 0.46 cm$^{-1}$, at 532 nm wavelength, respectively[37]. See Supplementary Note 6 for the detailed Monte Carlo simulation setup[38]. Due to the light diffusion and scattering, the light fluence uniformity quickly developed with the penetration depth. The region with a light intensity drop over 20 dB

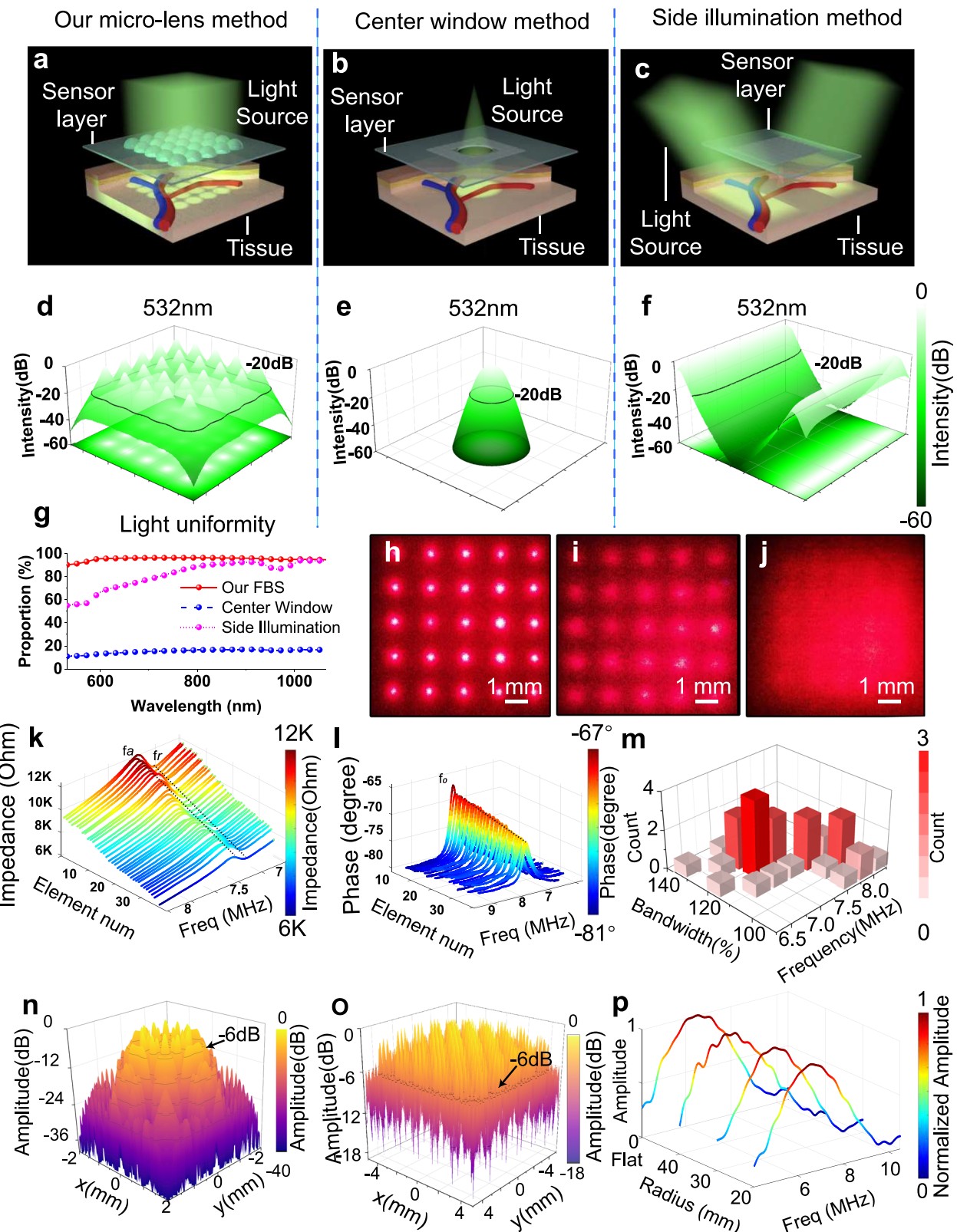

below the maximum is defined as the blind region at a specific depth to facilitate quantitative analysis. At the depths of 10 mm, all three illumination solutions have uniformly covered the region of interest without blind regions. Nevertheless, neither the side illumination method nor the center window method could achieve a distribution uniformity at the superficial zone: at a depth of 1 mm (the typical depth for some superficial veins), both side illumination and center window

methods have a large blind region under 532 nm light illumination (Fig. 2e, f); hence a light pre-diffusion layer is on-demand in these methods, which inevitably increases the device thickness[36]. In contrast, our micro-lens array with multi-focusing capability brings about a relatively high light distribution uniformity, with the blind region only laying at the four corners, as shown in Fig. 2d. A similar comparison of light illumination patterns at 1000 nm wavelength can be found in

**Fig. 2 | The design and characterization of optical and acoustic elements.**
**a**–**c** Schematic diagram of Monte Carlo simulation under different optical illumination settings. Our micro-lens array method (**a**) is compared with other classical methods, including the center window method (**b**) and the side illumination method (**c**). **d**, **e** The 532 nm light fluence distributions after penetrating 1 mm tissue. Our micro-lens array method (**d**) shows a larger effective illumination area (determined by −20 dB illumination intensity) than the center window method (**e**) and side illumination method (**f**). **g** The proportion of effective illumination area at 1 mm tissue depth with different laser wavelengths. Red for our OBS, blue for the center window, and pink for side illumination. **h**–**j** Photographs of illumination pattern of 633 nm laser after passing through the OBS (**h**), the OBS with one layer of paper at bottom (**i**), and the OBS with two layers of paper at bottom (**j**). **k** Impedances of 36 acoustic sensors (sorted by descending order). **l** Phases of 36 acoustic sensors (sorted by descending order). **m** The histogram of the bandwidths and center frequencies of 36 acoustic sensors. **n** The receiving field intensity of a single acoustic sensor at 1 mm depth. **o** The acoustic field of view (AFOV) of 36 sensor elements at 1 mm depth. **p** The response spectra of acoustic sensors under different bending conditions. Source data are provided as a Source Data file.

Supplementary Fig. 4, and the percentage of the effective illumination areas at different depths for 532 nm and 1000 nm lights are illustrated in Supplementary Fig. 5. Our micro-lens array shows the highest effective light illumination area at various depths among the designs. For the broad spectrum from 532−1064 nm at 1 mm depth (Fig. 2g), the effective illumination area exceeds 90%, proving that our micro-lens array can realize more uniform and larger area light delivery than classical methods. Therefore, no additional pre-diffusion layer is required, and the device can be fabricated into a thin film. To experimentally demonstrate the excellent light uniformity of the micro-lens array on our OBS, a beam of light (633 nm) irradiated through the OBS (Fig. 2h), with one layer (0.2 mm) (Fig. 2i) and two layers of paper (0.4 mm), as the optical diffusion medium, overlayed from the undersurface (Fig. 2j), respectively (Supplementary Note 7). Meanwhile, the effects of light patterns under different bending situations are also studied (Supplementary Note 8). In all cases, the transmitted light patterns show a uniform distribution behind a thin diffusion layer. Apart from this, a phantom absorption experiment verified the OBS's capability for multi-spectral imaging (Supplementary Note 9).

The design principle of acoustic elements, which resembles the general rule of the ultrasound sensor array, includes evaluating the electromechanical characteristics (resonance and antiresonance frequencies), acoustic resolution, and acoustic field of view (AFOV). The mean resonance frequency $f_r$ and antiresonance frequency $f_a$ (Fig. 2k) are 7.64 MHz and 8.01 MHz with s.d. (standard deviation) of 38.4 kHz and 39.2 kHz, respectively. The center frequencies of the acoustic sensors average 7.8 MHz (s.d. = 37.7 kHz) according to the phase angle spectra (Fig. 2l). The stable electromechanical performances indicate good uniformity among the acoustic elements produced from a reliable fabrication process. The thickness of the PVDF used is 110 μm with a bare resonance frequency of 10 MHz, which is decreased with the coating layer of silver electrodes. A thinner silver electrode layer favors higher resonance frequency but may result in excess electrical signal loss during signal transmission[39]. A multi-layered Mason's model (Supplementary Note 10) allows for pre-evaluation of the electrical impedance and phase angle, enabling prefabricated optimization with the fine-tuned thickness of the electrode as well as other matching and backing layers. After integrating the acoustic element into the OBS system, the acoustic element's impulse responses could be characterized by a point-like black tape target. As shown in Fig. 2m, most of the center frequencies peak ~7-8 MHz, and ~78% of sensors have a bandwidth of over 100%. The corresponding attainable axis resolution can be worked out to be 200 μm in tissues with a sound speed of 1500 m/s (Supplementary Note 11), and the lateral resolution is ~1200 μm at different depths after image reconstruction (Supplementary Note 18).

Similar to the criteria for optical illumination, the AFOV uniformity within the whole ROI should also be characterized (Supplementary Note 11), and the AFOV tends to be uniform along with the increase of detection depth. The uniformity of AFOV at the depth of 10 mm with an s.d of only 1.5 dB; in contrast, the shallow layer may suffer serious uniformity problems. The receiving beam pattern of a single sensor at a depth of 1 mm is illustrated in Fig. 2n. The simulated effective spatial response region (−12 dB) swiftly exceeds the sensor area of 2.28 mm² and reaches 5.5 mm² (3.17 mm² from measurement). Profited from the small gaps (0.15 mm) between acoustic sensors, the superimposed AFOV of the whole OBS at depth of 1 mm reveals a superb uniformity (Fig. 2o); the simulated mean versus maximum acoustic amplitudes from the effective AFOV is about −6.5 dB (−3.1 dB from measurement) with an s.d. of 2.3 dB (1.4 dB from measurement) which can be further reduced to 1 dB at a depth of 2 mm. The actual acoustic field is characterized by a phantom experiment (Supplementary Note 12). Meanwhile, the minimized inter-element gap size also avoids the grating lobe-induced artifacts (Supplementary Note 13). In addition to simulation, a bending test was conducted by sticking the sensors firmly onto different rigid curved surfaces (with a radius of 40 mm, 30 mm, and 20 mm, respectively) to compare the spectra of the received signal (refer to Supplementary Note 8 for the setup). The spectral results (Fig. 2p) suggest that the OBS can substantially maintain its receiving performance under these bending conditions. Even after being irradiated by the laser 24,000 times in 8 days, the performance of OBS is not compromised at all (Supplementary Fig. 6).

## 3D imaging algorithm

The received raw data from OBS should be processed with the image reconstruction methods before the blood physiological information extraction. The classical reconstruction method, based on the delay-and-sum (DAS) principle and implemented in time-space domain[40], is not suitable for flexible sensing and imaging in terms of both accuracy and efficiency. The fundamental misconception of the DAS-based method is that the skin should be considered as a heterogeneous layered medium rather than a simple homogenous component. Each layer possesses different acoustic properties that cause a serpentine receiving path (Fig. 3a), making it challenging to evaluate DAS's delay values accurately. Moreover, as the acoustic element is bent and tightly attached to the non-planar skin surface, the path calculation will become even more complicated due to inter-sensor displacement. Also, to enhance the resolution of the blood vessels 3D structures, several interpolations are required before rendering at a substantial computational cost in DAS-based reconstructions. Recently, deep-learning-based methods have been reported to overcome these challenges[41,42]. Generally, these solutions demand a vast amount of data, which highly determines the model performance. Here, we propose a phase shift non-uniform Fast Fourier transform (PS-NUFFT) algorithm to reconstruct the 3D image in a general heterogeneous medium setup. This method decomposes the wave components and utilizes wavefield extrapolation to separately examine the propagation of each wave component (Fig. 3b), which avoids the calculation of the complicated receiving path. Moreover, pre-migration technique[43] could be integrated into the PS-NUFFT algorithm to adapt to the curved acoustic elements distribution problem. PS-NUFFT is established upon the more exact Fourier domain physical modeling which could account for various acoustic effects (Supplementary Note 14). Thanks to its highly low-cost computational memory and complexity, the PS-NUFFT is eminently suited for our lightweight OBS sensor.

A numerical simulation was run to verify the performance of the PS-NUFFT. The detailed simulation setup is described in

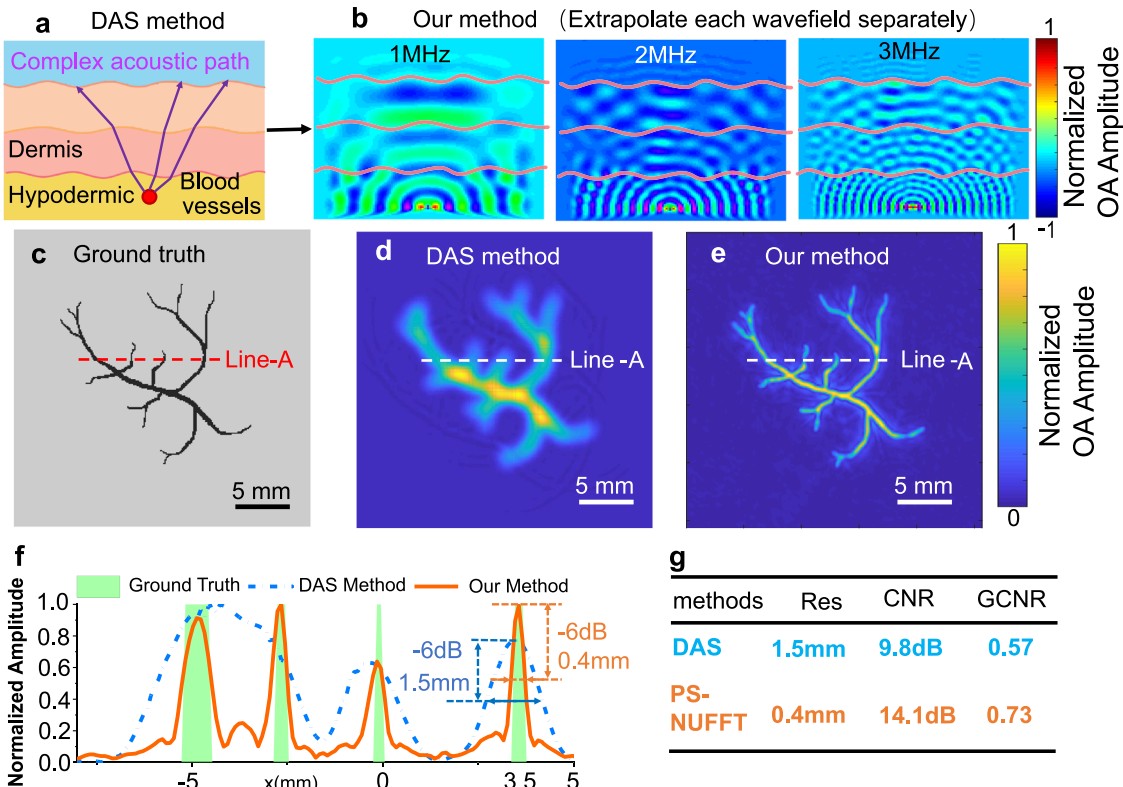

**Fig. 3 | 3D imaging algorithm. a** Schematic illustration of ultrasound receiving paths in the traditional Delay-and-Sum (DAS) method. A correct path takes account of refraction at the layer interface, and the time flight becomes complicated to calculate. **b** Our method decomposes the wave components and separately extrapolates the wavefields in different layers for image reconstruction. **c** The phantom image for numerical simulation. **d** Image reconstructed by traditional DAS method. **e** Image reconstructed by our method. **f** The amplitudes profiles, along with Line-A data of the ground truth (green band) and the images reconstructed by the DAS (dash blue) and our methods (solid orange), respectively. With the full-width half maximum (−6 dB) used as the resolution metric, our method outperforms the DAS method by 308%. **g** Quantitative comparisons between the DAS and our method in terms of resolution, CNR, and GCNR. Source data are provided as a Source Data file.

Supplementary Note 15, and the acoustic source pattern is depicted in Fig. 3c. The images reconstructed by the DAS-based and our PS-NUFFT methods are shown in Fig. 3d, e. Clearly, PS-NUFFT reconstructed image shows a much higher resolution close to the sources. A quantitative comparison of the image resolution could be evaluated by slicing the data amplitude profile along Line-A for both images (Fig. 3f). Using full width at half maximum (−6 dB) as the resolution metric, the resolution of a point source at $x = 3.5$ mm is 0.4 mm for the PS-NUFFT method and 1.5 mm for the DAS method, indicating our method outperforms imaging finer structures. Additionally, the PS-NUFFT is superior to the DAS method with a better contrast-to-noise ratio (CNR, 14.1 dB vs. 9.8 dB) and generalized contrast-to-noise ratio (GCNR[44–47], 0.73 vs. 0.57.) (Fig. 3g). In completing such a 3D imaging task with a large data size of $256 \times 256 \times 300$ ($x \times y \times z$) pixels, our PS-NUFFT only consumes about 2 seconds on the image reconstruction with a light-weight laptop (ASUS 430UN, CPU8550U @ 1.8 GHz). In comparison, DAS spends 2 s only to obtain one piece of depth information and would cost nearly 600 seconds on the whole data set. For low-density OBS arrays, PS-NUFFT also significantly outperforms DAS in terms of SNR and resolution consistently across different settings (Supplementary Note 16). Meanwhile, by integrating the pre-migration method[43], PS-NUFFT could compensate for the limited bending-induced image distortion (Supplementary Note 17).

### Hypoxia and exogenous agent decay in mice

The OBS with optimized optical and acoustic coupling can characterize blood physiological information from measured OA signals at different depths (Supplementary Note 18), which was first verified with in-situ hypoxia monitoring in mice. The principal function of blood is to deliver oxygen from alveoli to tissues. Consequently, the pathological hypoxia, likely due to COVID-19, will cause physical dysfunction and even directly endanger human life and personal safety[48]. Therefore, monitoring hypoxia status should aid in the early detection and prevention of related physiological abnormalities. The hypoxia test was conducted with the OBS attached to the mice's abdomen skin (Fig. 4a). The mouse was anesthetized first and then supplied with two types of air-nitrogen-isoflurane mixture: A mixture dosage (air 2 L/min, nitrogen 0 L/min, isoflurane 2%) can provide a normal oxygen condition (21% $O_2$), and another proportion (air 0.5 L/min, nitrogen 1.5 L/min, isoflurane 2%) can set up an oxygen-poor condition (hypoxia, 5.25% $O_2$). With the two types of the mixture supplied alternatively, the OBS continuously monitored the blood status of the mice under different conditions. Under the oxygen-poor condition, hypoxia led to lower oxygen saturation. Each hypoxia cycle took about 230 seconds, comprising 50 seconds of an oxygen-poor state and 180 seconds of a normal oxygen state. Figure 4b provides a statistical result that the optoacoustic amplitudes recorded by OBS are strongly correlated to the oxygen saturation (measured by oximeter) with an $R^2$ of 0.99. Every 50% decrease in oxygen saturation causes about 38 mV reduction of optoacoustic amplitudes. In addition, our device can discriminate the hypoxia trend from both the veins and the arteries (while the pulse oximeter can only retrieve artery information). Figure 4c presents a set of hypoxia experiments on mice (the other two sets are shown in Supplementary Fig. 7), where it can be found that the OA amplitudes of the artery basically followed the trends of the oximeter, while the veins showed a noticeable delay at the appearance of falling and rising

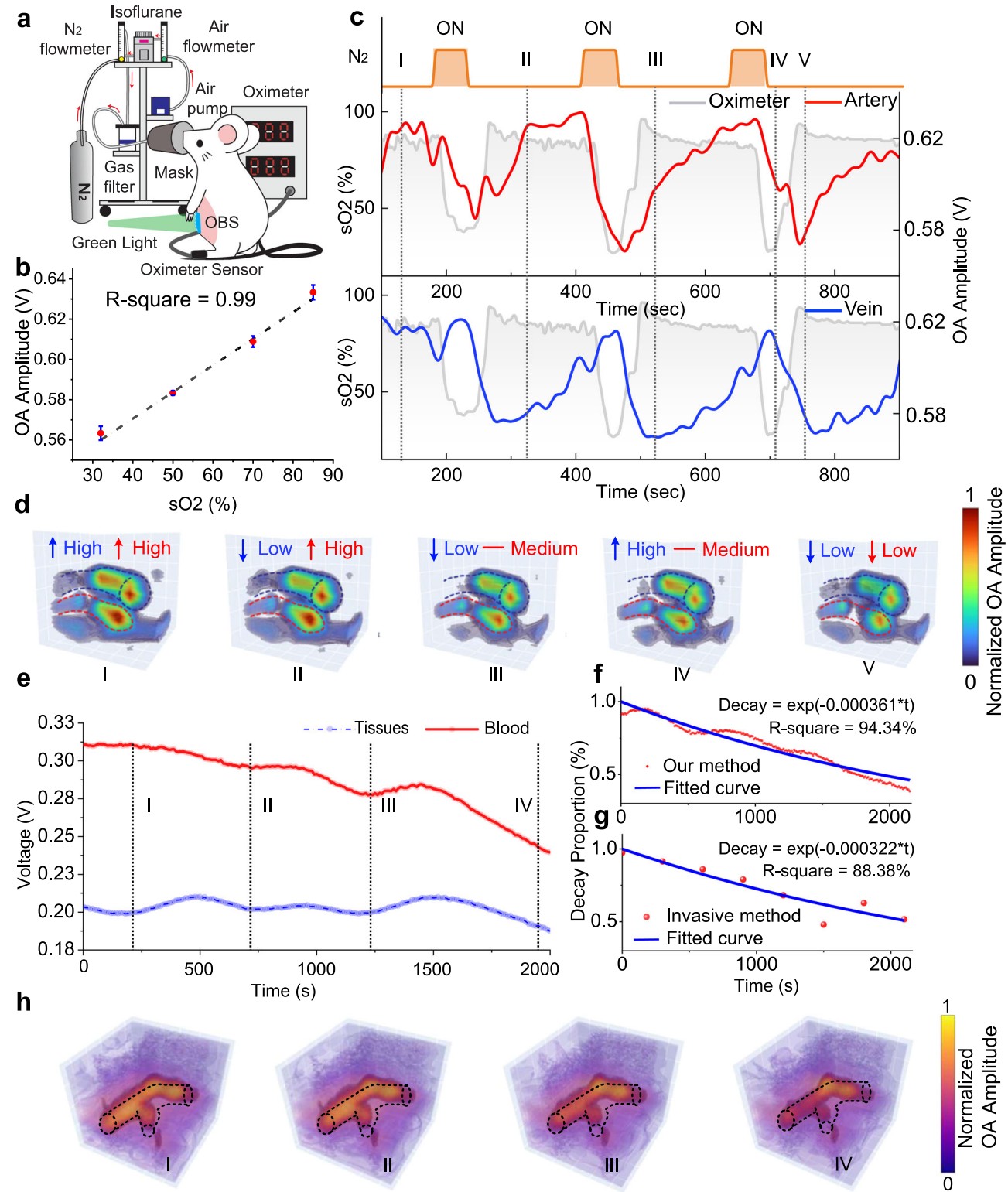

edges. According to ref. 49, different blood vessels respond differently to hypoxia, and venous blood often has a delayed response relative to arterial blood. Also, the dynamic oxygen changes of both venous and arterial blood are displayed with 3D imaging using OBS (Fig. 4d and Supplementary Movie 1).

Besides hypoxia status, the intravascular exogenous agent (Supplementary Note 19) concentration decay can also be monitored with the OBS, like Rhodamine B (Supplementary Note 20). Tracking the exogenous agent concentration in the bloodstream demonstrates its

importance in pharmacokinetics and pharmacodynamic analysis, contributing to optimizing the prescription of medicine dosage, dosing interval, and the suppression of the toxic side effects[50]. Traditional intravascular exogenous agent concentration monitoring is based on the intermittent invasive blood test, while the OBS provides the feasibility of the approach for noninvasive and continuous monitoring of the decay. The mouse was anesthetized in the exogenous agent concentration tests and intravenously injected with 200 μL Rhodamine B (2 mg/mL) from the tail as a labeled exogenous agent. After

**Fig. 4 | Hypoxia and intravascular exogenous agent concentration decay tests on mice. a** The hypoxia experiment setup for in vivo animal tests. **b** The linear correlation between sO2 and OA amplitudes. **c** The trend lines of peak-to-peak values of the OA signals measured by OBS in the hypoxia test. The orange regions represent periods with nitrogen supply (low oxygen status). Blue and red lines are OA amplitudes of veins and arteries, respectively. The gray region represents the oxygen saturation measured by an oximeter. **d** Snapshot images of the reconstructed 3D mice blood oxygen mapping at 30 s, 320 s, 520 s, 710 s, and 760 s. **e** The trend lines of peak-to-peak values of OA signals measured by OBS for intravascular exogenous agent concentration monitoring. Rhodamine B metabolism in blood

vessels (red line) and the tissues (blue line) show the trend of decay over time. **f** The decay trend of Rhodamine B in blood (red line) measured from the OBS. The decay coefficient from the solid blue fitted curve is $3.6 \times 10^{-4}\,s^{-1}$, $R^2 = 94.3\%$, and RMS = 91.8%. **g** The decay trend of Rhodamine B in blood (red points) measured from the in vitro fluorescence method. The decay coefficient from the solid blue fitted curve is $3.22 \times 10^{-4}\,s^{-1}$, $R^2 = 88.4\%$, and RMS = 86.4%. **h** Snapshot images of 3D mice Rhodamine B concentration mapping at 200 s, 700 s, 1200 s, and 1900 s in concentration decay test. Rhodamine B in the blood is rendered orange, whilst Rhodamine B in tissues is rendered purple. Source data are provided as a Source Data file.

Rhodamine B is stably metabolized (about 1 h after injection), the OBS was attached for continuously monitoring the mouse abdomen region, providing continuous 3D imaging of mouse blood vessels (Supplementary Movie 2). The trend line of OA signals (Fig. 4e) captured a decay of Rhodamine B level in the blood, where the value was reduced by 28% at 2000 s. Meanwhile, during the metabolism, the exogenous agent was released from the blood into tissues. Such diffusion phenomenon could be visually discriminated from the photographs of the mouse abdomen region, taken before and after Rhodamine B injection (Supplementary Fig. 8); the mean intensity of the red color (RGB mode) was elevated from 143 to 156 (color scale 0–255) after injection. Generally, the decay rate of Rhodamine B in the tissues was much slower than that in the blood, and as a result, the trend line of tissue was reduced by only 10% at 2000 s. Considering the tissue signal as the baseline, the decay of Rhodamine B in the blood conformed to the exponential decay curve with a decay coefficient of $-–3.6 \times 10^{-4}\,s^{-1}$ and $R^2$ of 94.3% (Fig. 4f). For comparison, blood sampling was taken every 5 mins concurrently for the in vitro measurement of the Rhodamine B concentration with fluorescence tests (Supplementary Note 21). The decay coefficient of Rhodamine B obtained from in vitro test was $-3.2 \times 10^{-4}\,s^{-1}$ ($R^2 = 88.4\%$), which was comparable to the OBS result (Fig. 4g). It is worth noting that the OBS measurement provides a better $R^2$ and root mean square (RMS). Moreover, the decay of Rhodamine B was discernible from the snapshot images at sequential time points (Fig. 4h).

## Compliance for veins

A venous system nearly houses 70% of blood volume that significantly influences cardiac output by regulating blood flow back to the heart, which intensely relies on the mechanical properties of the venous wall. The compliance of the venous wall is a crucial indicator in the evaluation of cardiovascular status and in predicting venous diseases[51]. Some studies have shown that venous thrombus can increase the expression of MMP-9, a matrix-remodeling protein known for its ability to lyse both elastin and collagen. However, MMP-9 is also responsible for degrading the extracellular matrix and basement membrane of the venous wall leading to a decrease in wall compliance of veins[52]. In this work, before evaluating the properties of the venous wall, a set of blood perfusion tests was employed to explore the venous reactions during vascular occlusion and release. The venous perfusion experiment was targeted at the human dorsal venous network (Fig. 5a). After attaching the OBS to the dorsum of the hand (Fig. 5b), the vascular occlusion was initiated, and the complete experimental process was recorded in Supplementary Movie 3. The amplitudes of the trend line measured from the venous OA signals were increased by 23% during vascular occlusions. The main reason is that the inflation of the cuff worn on the arm blocked the return of blood, resulting in blood pooling in veins, which enhances optical absorption and hence the OA amplitudes. After releasing the pressure, the trend line rapidly dropped. A sequence of snapshot 3D images (Fig. 5d) visually displays the amplitude variation of blood signals for the venous volume

change's subsequential characterization. Such a phenomenon is cross-validated with ultrasound imaging (detail setup see Method) with a similar trend line denoting the evolution of the cross-sectional area of the vein (Supplementary Fig. 9 and Supplementary Movie 4).

A further quantitative experiment was conducted to explore the venous volume change under different occlusion pressures. For inducing adequate vasodilation, the cuff was worn at the upper arm upright of the subjects ($n = 3$) as described in Method. The occlusion pressure was adjusted from 110 mmHg to 0 mmHg with several step releases (30 mmHg decrement in the first three releases and 20 mmHg for the last decrement) (Fig. 5e). According to ref. 53, the typical venous pressure of the dorsal hand vein was ~30 mmHg for the upright sitting posture (Supplementary Fig. 10). To avoid complete flattening of the blood vessels during measurement, only amplitude ratios recorded at occlusion pressure of >40 mmHg were considered valid; the amplitude ratios 102%, 108%, and 114% were obtained from the conditions where the occlusion pressures are 50 mmHg, 80 mmHg, 110 mmHg, respectively. Such a decreasing trend was consistent with the venous volume measured with ultrasound imaging but with reduced errors (Fig. 5f). Since the amplitudes of recorded acoustic signals were proportional to the cross-sectional area of the vein as mentioned in the perfusion test above, it could be used to estimate the vena cava volume versus transmural pressure[51,54]. The compliances of veins ($\Delta R/\Delta P$, $R$: amplitude ratio, $P$: pressure) were ~0.196% mmHg$^{-1}$ and 0.185% mmHg$^{-1}$ at ranges of 50–80 mmHg and 80–110 mmHg, respectively in our experiment. Measured with venous-occlusion plethysmography (VOP)[55] setup, venous compliance is prone to be overestimated; this is because the approach focuses not solely on venous but is also susceptible to the volume change in the other tissues and blood vessels, including the arterioles and capillaries. Ultrasound imaging is another alternative solution to directly observe the venous volume changes, but the complex proximal supporting skeleton and bucky setup are incompatible for long-term monitoring. Our OBS demonstrated its promise of becoming a paradigm-shift technology for venous disease care due to its positioning capability for specific veins.

## Flow-mediated dilation for artery

Human arterial hemodynamic studies have enormous implications for cardiovascular disease prediction and diagnosis. Diseases like atherosclerosis, stroke, and thrombosis could be directly or indirectly reflected in blood pressure, blood flow, and shear stress in hemodynamic analysis[56–58]. In this study, several cardiovascular-related parameters of the artery, including flow-mediated dilation (FMD), were measured with OBS. Firstly, an arterial perfusion test set was employed to study the arterial reactions in the radial artery (Fig. 6a, b). Contrary to the veins experiment, the amplitudes of arterial signals dropped by 37% during the vascular occlusion since the blood flows into the wrist was blocked (Supplementary Movie 5). Correspondingly, the blood flushed into the radial artery upon releasing the pressure. The sudden influx of blood immediately recovered the signal amplitudes and even above their baselines for a short period (Fig. 6c, d). The trend line was

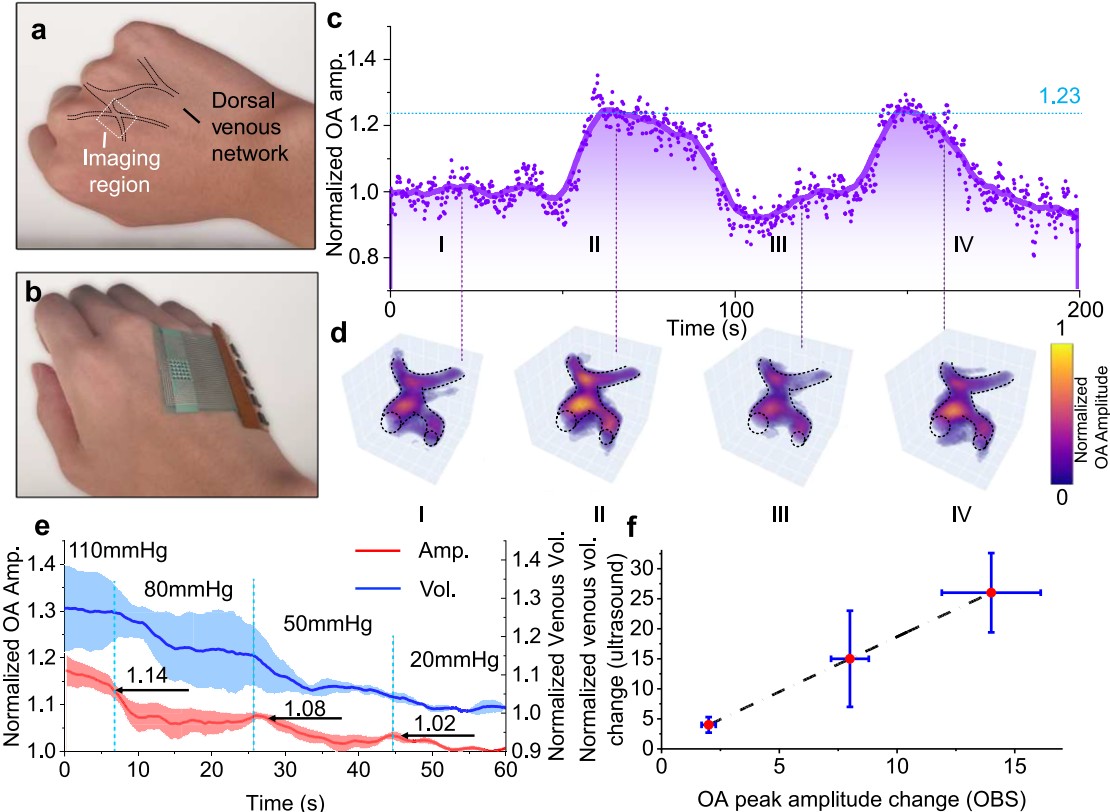

**Fig. 5 | Vascular occlusion tests on veins. a** Photograph of the outlined dorsal venous network, in which the image region is denoted with a white dotted box. **b** Photograph of the OBS worn on the hand. **c** The trend lines of peak-to-peak values of the OA signals from the OBS. **d** Snapshots of 3D vascular images at 20 s, 70 s, 120 s, and 170 s with the dashed line showing the venous network contour. **e** The comparison of peak amplitude and volume ratios under different pressures. Both the peak amplitudes and volumes decreased along with the reduction of occlusion pressure (Amp for amplitude, Vol for volume, data are presented as mean values ±s.d.). **f** the linear correlation between OBS-measured peak amplitude ratio changes and ultrasound-measured venous volume ratio changes under different cuff pressures ($n = 3$ independent experiments, data are presented as mean values ±s.d). Source data are provided as a Source Data file.

also further compared with medical ultrasound results, coinciding with a similar curve pattern (Supplementary Fig. 11 and Supplementary Movie 6).

Many studies have revealed that endothelial dysfunction is deleterious and implicated as a key factor in the initiation and progression of the atherogenic process that underlies coronary artery disease, peripheral ischemia, and some forms of stroke[59]. The FMD has been widely used as a medical standard to examine endothelial dysfunction for predicting cardiovascular risks[60]. Classical FMD tests utilize ultrasound imaging to continuously track the maximum diameter variations within one minute after the occlusion release. The low contrast of ultrasound and the high sensitivity to the measurement position heavily relies on bulky auxiliary fixtures of the ultrasound system to eliminate the motion-induced measurement error[61] (Supplementary Fig. 12). Taking advantage of the OA effect, our OBS simplifies the system integration while keeping pertinent specificity. Figure 6e–g shows the typical evolution of OA signals during the FMD tests ($n = 3$), where Peak A denotes the maximum vascular volume within one minute after the release of the cuff. The relative signal changes $p_A$ were +16.3%, +15.5%, and +9.3% at Peak A, respectively. Since the signal amplitudes are approximately proportional to the vascular volume changing in a small range, the OBS estimated vascular diameter changes ($\sqrt{(1+p_A)} - 1$) were 7.8%, 7.2%, and 4.6%, which were also consistent with the ultrasound imaging result of 8%, 7.1% and 5.6% in Fig. 6h (Supplementary Fig. 13 and Supplementary Movie 7). Note that a typical FMD value is around 6.7% for healthy people in the clinic, and every 1% decrease in FMD implicates a 12% increase in the hazard ratio for the cardiovascular system[62].

## Discussion

We have reported a flexible optoacoustic blood 'stethoscope' for multiparametric cardiovascular healthcare and verified its applications in continuous monitoring of hypoxia, intravascular exogenous agent concentration decay, and cardiovascular-related critical hemodynamical parameters. Taking advantage of optical spectral specificities and high acoustic resolution, the OBS allows imaging to quantitatively analyze multiple physiological vitals of blood to potentially solve several unmet clinical needs in the future. Besides, the flexibility and bendability enable the device to attach firmly to human skin for physiological monitoring. All the presented physiological vitals measurements are based on self-referenced relative percentage variation, rather than the change in the absolute OA amplitude whose baseline reading is vulnerable to skin tones and other complex in-vivo factors (Supplementary Note 22). 3D imaging provides a new perspective for the quantitative and specific studies of the prediction of cardiovascular disease.

The OBS reported here opens up a new sensing dimension for existing flexible electronics. The detection range is extended from areas directly below the sensor patch to a much wider window, and the detection mode evolves from 1D sensing to 3D imaging. Meanwhile, the applications of the OBS are not only confined to the aspects discussed in this paper. By selecting a specific or combination of laser wavelengths, the probe can also be potentially extended to measure pH values[63], blood glucose[64], blood temperature[65], and even monitor the treatment of tumors[66]. Moreover, multi-wavelength measurement provides rich characteristic parameters to decouple the multi-variable estimation. In other words, the OBS has the potential to monitor and

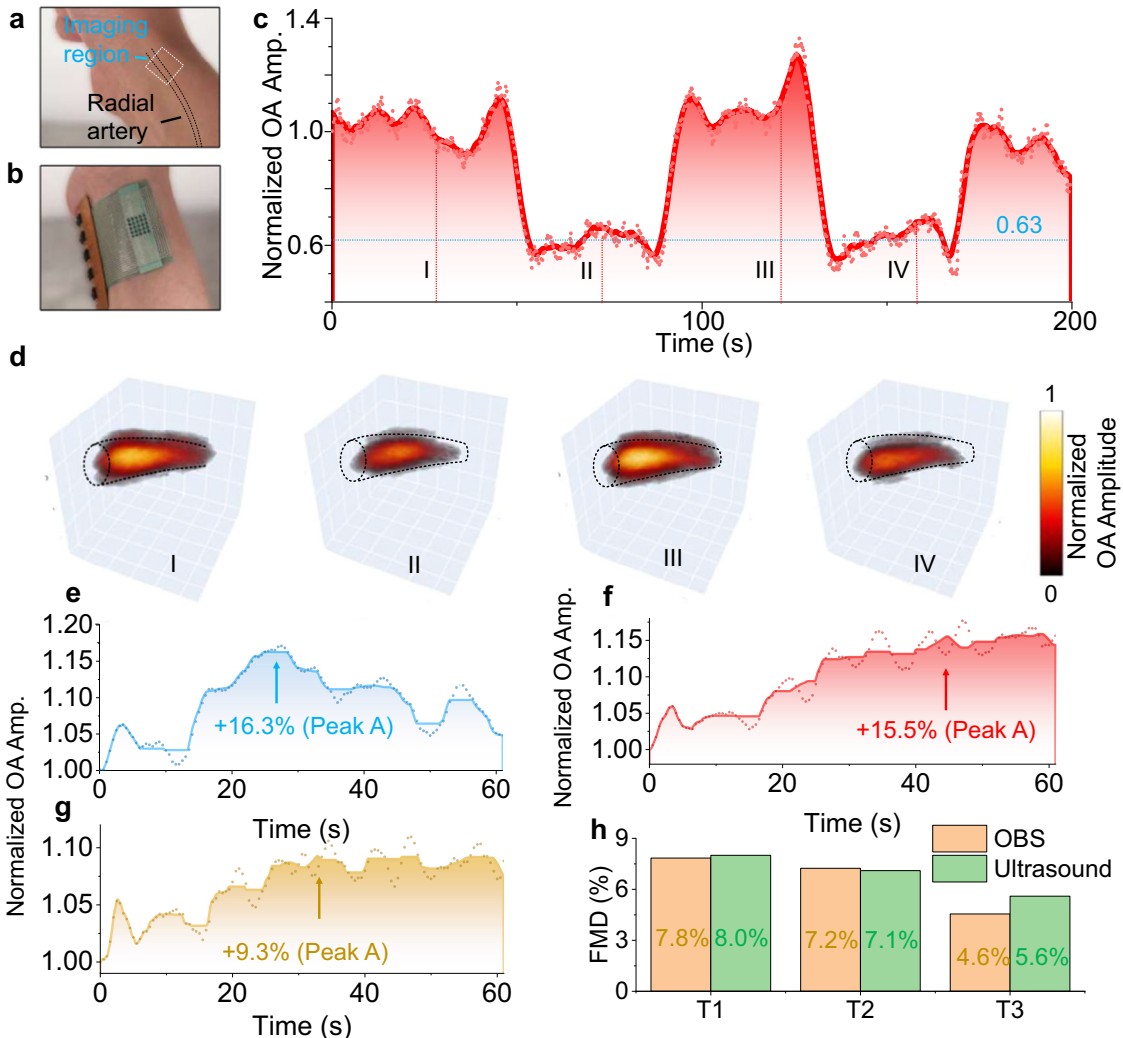

**Fig. 6 | Vascular occlusion tests on arteries. a** Photograph of the outlined radial artery, in which the image region is denoted with a white dotted box. **b** Photograph of the OBS worn on the wrist. **c** The trend lines of peak-to-peak values of recorded signals. **d** Snapshots of 3D blood images at 20 s, 70 s, 120 s, and 160 s with the dashed line showing the artery vessel contour. **e–g** The FMD tests with OBS. All the signals are recorded within 60 seconds after occlusion was released. Peak A represents the maximum vascular volume within one minute after releasing the cuff. **h** A quantitative comparison of our OBS and ultrasound imaging in measuring flow-mediated dilation (FMD). Orange for OBS measurement and green for ultrasound validation. Source data are provided as a Source Data file.

analyze multiple blood parameters simultaneously. Technical challenges still remain before deploying this preliminary prototype to other general settings. For example, higher-resolution imaging still requires denser acoustic and optical elements to increase imaging voxels. Additionally, limited to the current laser technology, the size of the laser driver is still unneglectable, and the front-end sensor patch yet requires a wired data acquisition system for post-processing. Future studies will focus on further improvement by integrating denser acoustic and optical elements, incorporating LED light sources, compacting acquisition circuits, and wireless transmission systems based on system-on-a-programmable-chip, for distributed, mobile, and real-time monitoring of animals and humans from various aspects. Also, further clinical validation may broaden the applications of the OBS to accelerate the development of mobile medical, as well as to advance current state-of-the-art clinical practices for the prevention and investigation of cardiovascular-related diseases.

## Methods

All animal care and procedures were conducted in strict adherence to the protocols sanctioned by the Institutional Animal Care and Use Committee (IACUC- A20033) of Nanyang Technological University,

complying with all federal guidelines. The mice were housed under conditions of an ambient temperature between 18–26 °C, relative humidity of 30–70%, and illumination of ~325 lux, positioned 1.0 meter above the floor for a 12-h light/dark cycle each day. Similarly, all protocols for human trials received approval from the Institutional Review Board (IRB-2021-02-028) of Nanyang Technological University, in line with federally approved guidelines. The experiments incorporated the participation of three healthy individuals (20–30 years old, 1 female, and 2 males) who gave informed consent. Age and sex were not considered as factors in this study due to their irrelevance to the experimental outcomes.

### Fabrication of the flexible optoacoustic blood 'stethoscope'

The piezoelectric acoustic element was fabricated from a 110 μm-thick polyvinylidene difluoride (PVDF) film (Measurements Specialties Inc, TE Connectivity Ltd). After cutting the PVDF film into the desired size, the thin films were rinsed in ethanol in the ultrasonic bath for 20 min and subsequently rinsed in DI water for cleaning. After the PVDF film was thoroughly dried in the air, the first layer of top electrodes was screen-printed with a 7 μm silver paste layer (Qingdao Poten Technology Co., Ltd) patterned by a shadow mask (designed by AutoCAD

software). Then a UV curable dielectric ink (Jelcon IN-15M, JUJO Chemical Co, LTD) was screen-printed on the top layer as insulation and cured with 120 W/cm UV lamps for 10 mins. The second layer of the top electrodes, which serves as the output wiring for later measurements, was screen-printed on top of the insulation layer with 10 μm silver paste. Following the top layer fabrication, a bottom electrode layer of 7 μm silver was deposited with screen printing. The pre-drilled holes on the acoustic sensor were pattern-cut with an electronic cutting machine (Silhouette Cameo).

The micro-lens array was manufactured by casting a mixture of PDMS elastomer and crosslinker (Sylgard 184) in a 10:1(w/w) ratio in the micromachining-fabricated mold (Computer numerical control) and then spin-coated at 250 rpm/min (EZ4 Spin Coater, Lebo Science). After that, the mixture was degassed in a vacuum oven and cured at 60 °C for 6 h. Next, the micro-lens array was peeled off from the mold. After that, the micro-lens array was mounted on top of the as-fabricated acoustic sensor array and secured with the deposition of PDMS precursor over its edge. Two vertical interconnect access (VIA) holes filled with conductive silver paste on each side of the device are utilized to route the bottom electrodes to the top plane before the final PDMS encapsulation.

## PS-NUFFT reconstruction process

In summary, the whole reconstruction process involves four main steps. First, the received raw data is converted into a wavenumber domain with a 3D Fast Fourier transform as

$$P\left(k_x, k_y, z = 0, \omega\right) = \frac{1}{8\pi^3} \int\int\int p(x, y, z = 0, t) e^{-ik_x x} e^{-ik_y y} e^{i\omega t} dx dy dt, \tag{1}$$

where $p(x, y, z = 0, t)$ is the pressure received by the acoustic element. Subsequently, the phase shift is used to extrapolate the wavefield onto the boundary of each layer

$$P\left(k_x, k_y, z = z_L, \omega\right) = P\left(k_x, k_y, z = z_L, \omega\right) \prod_{l=1}^{L-1} e^{ik_z(d_l)}, z_L$$

$$= \sum_{l=1}^{L-1} d_l, l = 1, 2, 3 \dots L - 1, \tag{2}$$

where $d_l$ represents the thickness of layer $l$. Then an integral transformation is implemented on the extrapolated wavefields with a non-uniform interpolation expressed as

$$P_I\left(k_x, k_y, k_z\right) dk_x dk_y dk_z = P\left(k_x, k_y, z = z_L, \omega\left(k_x, k_y, k_z\right)\right) ck_z / \sqrt{k_x^2 + k_y^2 + k_z^2} \Phi D_p dk_x dk_y d\omega, \tag{3}$$

where ϕ is a sinc interpolation basis, $D_p$ is a down-sampling function, and $c$ is the sound speed of tissue. Finally, the 3D image is obtained by a 3D inverse Fast Fourier transform on $P_I(k_x, k_y, k_z)$ as

$$S(x, y, z) = \int\int\int P_I\left(k_x, k_y, k_z\right) e^{ik_x x} e^{ik_y y} e^{ik_z z} dk_x dk_y dk_z, z \in [z_l, z_{l+1}], \tag{4}$$

where $S(x, y, z)$ represents the acoustic sources excited by the laser in the $l$-th layer. The completed acoustic sources are obtained by combining all the $S(x, y, z)$ in different layers. The total computational complexity for 3D imaging is $O(Ln^3 \log_2(n^3))$, where $L$ represents the total layer number. Since the number of the skin layers usually does not exceed 5, the overall computational complexity reduces below that of the DAS-based solution ($O(n^5)$) while achieving a much higher reconstruction resolution and accuracy in a compatible heterogeneous setup. Generally, interpolation is performed to smooth out the sawtooth effect in low-resolution images. In PS-NUFFT, the final image can be conveniently interpolated by zero-padding the spectrum of $k_x$ and $k_y$ or controlling the down-sampling function $D_p$ prior to the 3D inverse Fast Fourier transform; nevertheless, such operation barely imposes an additional computational burden.

## Experimental test platform

The test platform contains two parts, including the front-end probe and back-end electronics and optics (Supplementary Fig. 14). The front-end comprises the OBS and a flexible two-stage preamplifier, which can be attached to the region of interest on the skin. The light illumination is generated by a pulse laser (Radiant 532LD, Opotek Ltd, UK) and delivered via a bespoke side-firing fiber bundle. The laser energy is strictly controlled below 10 mJ, and the illumination fluence on the skin is limited under the safety level of 20 mJ/cm[267]. All the electrodes of acoustic nodes are connected to the customized on-skin two-stage multi-channel preamplifier (MAX4805A, maxim integrated, and TLV3544, Texas Instruments) flexible board by anisotropic conductive film tape. The preamplifier was fabricated on a flexible substrate for directly bonding to the OBS without introducing any interconnection cable to minimize the possibly additional imposed interference and signal loss. Then, the data is split for the A-scan display and 3D imaging reconstruction separately. The A-scan dataflow subsequently passes through multiplexers (TMUX1108, Texas Instruments), low noise amplifier (5072PR, Olympus), oscilloscope (Lecroy), and finally, it is transferred to a laptop via Ethernet for direct display. The 3D imaging raw data are directly amplified and recorded by a multi-channel DAQ (SonixDAQ). After that, the data were sent to the laptop via USB and then processed with the PS-NUFFT reconstruction method, and it was finally post-rendered for the final 3D display with a frame rate of 10 Hz (mainly restricted by the light source pulse repetition rate). A Xilinx FPGA controller (MiniZed, AVENT) is implemented (Vivado version 2017.4) to control all the data acquisition processing. Before attaching the OBS, a magnetic marking pen (Ascension 3D Guidance drive BAY) is used to register the skin surface profile coordinates. Thus the positions of each acoustic node are determined to accommodate the skin curvature in the image reconstruction (Supplementary Note 23).

## Animal procedures

Four Nude mice (CrTac: NCr-Foxn1nu, Male, 6–8 weeks, Invivos) were involved in oxygen saturation and exogenous agent concentration decay tests. The mouse was fixed in an animal holder with a gas-supply apparatus mounted on the head. The apparatus was supplied by the anesthesia machine (R540IE, BIOSEB) with a mixture of air, nitrogen, and isoflurane (Supplementary Fig. 15). The concentration of oxygen inhaled was controlled by adjusting the flow rate ratio of air and nitrogen. For comparison, a mice oximeter (MouseSTAT Jr. Pulse Oximeter) is clamped on the paw of mice to record the arterial oxygen saturation. A heating pad was placed under the mouse to maintain its body temperature, and the ultrasound gel was applied to ensure efficient acoustic coupling between the skin and the OBS receiver. During the trials, strict 'control of variables' scheme (with a customized integrated physiological monitoring system featuring pulse oximetry and temperature modules, etc.) was applied to avoid influence from unwanted sources. The accuracy and specificity of all the presented single biomarker measurements were evaluated against respective gold-standard methods.

## Human trial protocols

**Perfusion experiment.** The volunteers sat on the chair with a cuff worn on the upper arm. Before applying vascular occlusion, ultrasound imaging was acquired to locate the blood vessels. The arteries can be easily distinguished from the veins due to the periodic vasodilation-induced motion. Then, the OBS was

attached to the desired region for imaging. Our perfusion protocol involves the following five steps: (a) volunteers seated and rested for 1 min, (b) the pressure cuff was inflated to 150 mmHg and kept for 40 seconds to induce vascular occlusion, (c) the cuff was released entirely and held for 40 seconds, (d) the 150 mm pressure was again applied for another 40 seconds, (e) and at last, the pressure was again utterly released.

**Venous distensibility.** During the venous distensibility test, the initial venous pressure was maintained at ~30 mmHg to prevent complete flattening of the blood vessel. Since the venous pressure was directly determined by body posture, volunteers were required to bend their elbows at 90 degrees to achieve an ~40 cm difference in height between the testing area and neck (Supplementary Fig. 10). The test was started by inflating the pressure cuff to 110 mmHg to produce venous-occlusion. Then, the pressure was step-released from 110 mmHg to 0 mmHg with each step of 30 mmHg. The time interval between each release was about 20 seconds.

**Flow-mediated dilation.** The FMD measurement adopted the distal testing setting with a cuff worn on the upper arm. Simultaneously, the OBS was attached to the lower end wrist to characterize the FMD of the radial artery. In the beginning, 160 mmHg of total occlusion pressure was applied for 2 mins (cuff-on period: the gross occlusion pressure induced by a cuff equals the systolic blood pressure plus 50 mmHg) before the cuff was released. Then the data recording only started after the signal amplitudes first time recovered to their baseline. A classical FMD indicator estimates the relative ratio of increment in hyperemia status artery diameter to baseline artery diameter.

## Ultrasound imaging for cross-validation

The medical ultrasound system (SonixRP, Ultrasonix, Canada) with a linear array transducer (L14-5/38) was used to monitor the veins and arteries dynamically. The image acquisition frame rate of the medical ultrasound machine was 55 Hz. The images were pre-processed with morphological methods to estimate the vascular cross-sectional areas (Supplementary note 24).

## Statistics and reproducibility

Plots were presented as mean ±s.d. with individual data points and experiments were repeated at least 3 times unless indicated otherwise. The experiments involved four nude mice and three consenting human participants. All collected data were meticulously analyzed using MATLAB, with no exclusions made in the data analysis process.

## Reporting summary

Further information on research design is available in the Nature Portfolio Reporting Summary linked to this article.

## Data availability

The main data supporting the results of this study are available within the paper and its Supplementary Information. Any additional requests for information can be directed to, and will be fulfilled by, the lead contact. The data generated in this study are available from Figshare at https://doi.org/10.6084/m9.figshare.23541531. Source data are provided in this paper.

## Code availability

The codes for the main algorithm (three-dimensional optoacoustic image reconstruction) within this paper are available at Zenodo repository[68] under accession code: https://doi.org/10.5281/zenodo.8051915.

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

## Acknowledgements

This research is supported by the Ministry of Education, Singapore, under its MOE ARF Tier 2 (Award no. MOE2019-T2-2-179).

## Author contributions

H.J., Z.Z., and Y.Z. designed the study. H.J. performed theoretical analysis, device design and interpreted the data. Z.Z. performed in vivo mice and human experiments. Z.C., Y.J., G.C., and J.W. fabricated and upgraded the device. Z.W. performed the mice experiments. C.Y. and W.S. developed the data acquisition system. H.J., Z.Z, W.L., Z.C., X.C., and Y.Z. composed the paper. All authors discussed the results and commented on the manuscript. Y.Z. supervised the work.

## Competing interests

The authors declare no competing interests.
