## [Peer review file · Nature Communications]

REVIEWER COMMENTS

Reviewer #1 (Remarks to the Author):

The authors reported a novel flexible photoacoustic sensor for wearable applications. They demonstrated the 3D imaging capabilities by testing human and mouse *in vivo*. This device could be valuable for long-term monitoring of blood vessels or biomolecules. The reviewer recommends accepting the manuscript after addressing the following concerns.

1. One recent paper also reported a wearable and flexible photoacoustic sensor, 'Gao, Xiaoxiang, et al. A photoacoustic patch for three-dimensional imaging of hemoglobin and core temperature. Nature Communications 13.1 (2022): 7757.'. The reviewer suggests the authors to discuss about the advantages and disadvantages of both devices.

2. The flexibility improves the wearability of the sensor. But it may also introduce some other problems, such as uncertain transducer element positions. What are the imaging results when the sensor is bent? The authors may also elaborate about ways to decrease this influence.

3. How is the device attached to the skin? PVDF membrane is flexible, but not soft enough to conform to the skin. Body motion may introduce the movement of sensor, and air gaps between the sensor and the skin. Did authors observe any motion artefacts during measurement caused by air gaps or other factors?

4. In fig. 4h, the color bar shows the blood in the range of 0~1 and tissue in the range of -1~0. What is the criterion? It is not clear why blood signal is higher than 0 but tissue signal is lower than 0 either. How to determine the boundary between the blood and tissue?

Reviewer #2 (Remarks to the Author):

The authors have addressed most of the concerns that were raised and the work is novel and interesting with sound methodology. I also appreciate the size comparisons to other systems. There is however one discrepancy between the response document and the manuscript regarding the gCNR references. The response document says that the additional helpful references were included, but I could not find them in the manuscript. I read the missing references, and I agree they are helpful as stated in the response document.

Reviewer #3 (Remarks to the Author):

This paper does not address a real problem as claimed. My review and decision remains as before when I reviewed this paper for Nature Biomedical Engineering.

Reviewer #1

The authors reported a novel flexible photoacoustic sensor for wearable applications. They demonstrated the 3D imaging capabilities by testing human and mouse in vivo. This device could be valuable for long-term monitoring of blood vessels or biomolecules. The reviewer recommends accepting the manuscript after addressing the following concerns.

We greatly appreciate your recognition of our work and the insightful feedback. We've taken the comments into account and have addressed each point as follows:

Q1: One recent paper also reported a wearable and flexible photoacoustic sensor, 'Gao, Xiaoxiang, et al. A photoacoustic patch for three-dimensional imaging of hemoglobin and core temperature. Nature Communications 13.1 (2022): 7757.'. The reviewer suggests the authors to discuss about the advantages and disadvantages of both devices.

Response: In response to the reviewer's comment, we present a comparison between our photoacoustic blood 'stethoscope' (OBS) and the photoacoustic patch reported in Gao et al. (referred to as PP for subsequent comparison). While both devices serve different purposes, with OBS measuring multiple cardiovascular biomarkers and PP focusing on temperature sensing and blood vessel occlusion, a technical comparison helps understand their unique features and potential improvements:

Overall structure:

OBS: flexible layered structure vs. PP: stretchable island-bridge structure

1. OBS features a flexible layered structure with a Polyvinylidene fluoride (PVDF) substrate, while PP employs a stretchable island-bridge structure with elastomer encapsulation and serpentine metal electrodes.
2. In the OBS, both the acoustic component, constituted by PVDF, and the optical component, featuring microlenses, exhibit flexibility. This stands in contrast to the PP which amalgamates rigid components such as Vertical-Cavity Surface-Emitting Laser (VCSEL) and Lead zirconate titanate (PZT).

Optical modules:

OBS: fibre-coupled pulsed laser with a microlens array vs. PP: VCSEL array integrated on the patch

1. The pulsed laser in our OBS does carry a higher cost relative to VCSELs in the PP. However, this cost could potentially be mitigated by employing a more specialized single-wavelength laser, instead of the currently used wavelength-tunable laser designed for proof-of-concept experiments.
2. The laser wavelength of our OBS is 532 nm while it is 850 nm in PP. The 850 nm laser in PP does offer commendable penetration and imaging depths (approximately 2 cm ex-vivo and 1 cm in-vivo), and the 532 nm laser employed in our OBS achieves a superior contrast-to-noise ratio when imaging blood, producing better signal quality and accuracy. This is because the 532 nm laser in our OBS has a more pronounced

optical absorption coefficient of blood versus background elements, such as water, lipids, and other absorbing components, than the 850 nm in PP. This results in enhanced signal quality and accuracy, particularly when imaging shallow blood vessels, where the 532 nm laser's absorption is less pronounced.

3. The imaging frame rate of OBS is higher than that of PP. Under normal operation, the OBS has a frame rate of 10 fps while the PP only has 1 fps. The OBS image' frame rate is strongly dependent on the Pulse Repetition Rate of the laser since no coherent averaging is required with high-intensity laser pulses. In contrast, the limited power of the VCSELs requires thousands of times of averaging, which increases data and processing burdens. Noted this frame rate comparison is approximate since the actual frame rate is also affected by various factors, e.g., skin tone, geometry and depth of blood vessel.
4. The OBS can image arteries while the PP cannot. No artery is presented in the PP image since the instant motion including the artery pulsation disrupts the detected signal coherence and vanishes the recorded amplitude after averaging. The OBS, in contrast, does not require averaging, thereby enabling more robust arterial imaging.
5. For future exploration of other applications, the light source may be replaced as its wavelength is the key to selectively targeting different biomolecules. OBS offers greater versatility for proof-of-concept and prototyping studies due to its replaceable optical module, while PP allows higher integration for a reduced footprint.
6. In addition to the shared thermal regulation issues in both devices, the integration of high-power electrical cables near the measurement sites in the PP device introduces additional potential safety concerns that need to be considered.

Piezoelectric materials:

OBS: PVDF vs. PP: PZT

1. The PZT in PP is rigid while the PVDF employed in OBS is flexible.
2. The imaging resolving capability is proportional to the bandwidth of the acoustic transducer. The PVDF transducer in the OBS peaks around 7-8 MHz with 100% bandwidth. In comparison, the rigid PZT composites in PP have a center frequency of 2.4 MHz and nearly 60% bandwidth; While its imaging depth may be superior, its resolution is inferior compared to that of the OBS.

The layout of elements and reconstruction algorithm:

OBS: even distribution, 5 × 5 microlenses, 6 × 6 transducer array vs. PP: uneven distribution, 4 × 6 VCSELs, 16 × 13 transducer array

1. Our OBS is flexible but not stretchable, the inter-element spacing is relatively fixed and hence the individual element position is easier to obtain as input for later-on image reconstruction to reduce errors. Whereas PP's stretchable polymer substrate introduces more complexity in determining patch surface topography.
2. The larger number of transducer elements in PP typically aids image reconstruction. Incorporating additional elements is technically feasible and part of our future plans to further improve this feature.
3. In OBS, both the optical and acoustic elements are evenly distributed, whilst the

distribution in PP is uneven. Since the VCSELs in PP take up some space but the laser used in OBS is pre-focused thus the through-holes are smaller than the acoustic node thus they can be evenly interlaced within the latter. Firstly, the uneven optical and acoustic fields may cause artefacts like grating lobes. Secondly, at shallow depths, blind spots are formed underneath the VCSELs where the acoustic field fails to cover.

4. OBS's frequency domain PS-NUFFT reconstruction offers superior accuracy and time efficiency compared to PP's time domain homogeneous Coherence-Factor-weighted-Delay-And-Sum (CFDAS) approach, which is elaborated in the manuscript.

This comparison aims to shed light on the unique strengths of both devices, recognizing that each offers unique advantages depending on the specific application requirements.

Q2: The flexibility improves the wearability of the sensor. But it may also introduce some other problems, such as uncertain transducer element positions. What are the imaging results when the sensor is bent? The authors may also elaborate about ways to decrease this influence.

Response: Thank you for raising this pertinent issue. The flexibility of the sensor indeed poses challenges, particularly the position change of the transducer elements upon bending, which can potentially influence imaging quality. Additionally, the PVDF elements' flexibility also necessitates the consideration of their bending effects.

We have devised the following strategies to mitigate these influences:

1. **Sensor Position and Shape Calibration:** To account for variances due to bending, we implement a calibration phase prior to each measurement. This involves mapping the human skin's profile using an Ascension 3D Guidance magnetic marking pen before attaching our probe. This method allows us to ascertain individual element positions relative to the skin's surface profile, as detailed in Supplementary Note 23:

"Before attaching the sensors to the dorsal hand, the skin surface curvature is measured with a magnetic marking pen (Ascension 3D Guidance drive BAY) for one-time calibration, therefore the skin surface information could be registered into coordinates. As depicted in Fig. N25, stereoviews can be created from measured point clouds; hence, the element positions in the ROI can be inferred. The skin surface profile can also be captured using any other stereo device, such as the LiDAR module on a smartphone or a stereo camera. With the development of microelectronics, it is promising that future LiDAR chips can be downsized and integrated into a flexible substrate to serve this function. When the OBS is attached to the limbs distant from joints and muscle groups, it is reasonable to assume that the surface is relatively smooth; as the skin bending around the forearm, dorsal hand, and thigh are all safely above 20 mm, which is the minimum radius considered in this work. Since OBS is not stretchable, the relative positions between acoustic elements are relatively immobilized.

Fig. N25. The stereoview of the forearm (a,b), dorsal hand (c,d) and thigh (e,f) skin profiles.”

Alternative techniques, such as those suggested by [1] and [2], include using an optical 3D scanner to acquire sensor profile data. However, these methods cannot track sensor profiles during measurement without restricting the sensor’s orientation or position relative to the scanner. In the future, the solution to the bending issue can be further improved by integrating the strain sensors within the device to track sensor profiles. This would enable the real-time monitoring of the sensor’s degree of bending, allowing for dynamic correction of any resultant imaging distortions with the reconstruction algorithm.

2. **Algorithmic Adjustment:** Under substantial bending, the assumption that individual elements are flat becomes invalid. To counter this, we leverage the derived sensor positions and shapes to apply a compensatory correction. Subsequently, we employ our previously developed algorithms, tailored for imaging on circular and irregular surfaces, to rectify these distortions [3]. This approach allows for the conversion of planar transducers into curved ones, enabling the application of ‘Virtual Detector Migration’. The simulation in Fig. R1 illustrates the image improvement achieved after implementing position and shape corrections. The parameters used for these simulations align with those outlined in Supplementary Note 16, which were chosen to reflect our actual setup. In this configuration, the ‘L’-shaped target within a single-layered media serves as the optoacoustic source. Optoacoustic waves are captured

by a 6×6 array of acoustic nodes, each of dimensions $1.6 \text{ mm} \times 1.6 \text{ mm}$. The bending radii assigned for the image pairs (b-c), (d-e), and (f-g) were 100 mm, 50 mm, and 20 mm, respectively. Evidently, as the degree of bending increases, distortions become more pronounced. However, the proposed calibration and compensation pipeline proves effective in mitigating these distortions.

Fig. R1 Simulation of reconstructed images before and after compensation. (a) Unbent condition. (b) Image at 100mm bending radius before compensation. (c) Image at 100mm bending radius after compensation. (d) Image at 50mm bending radius before compensation. (e) Image at 50mm bending radius after compensation. (f) Image at 20mm bending radius before compensation. (g) Image at 20mm bending radius after compensation.

In summary, we recognize the challenges posed by sensor flexibility, notably the accurate positioning of transducer elements when bent. However, we have addressed these through position and shape calibration and algorithmic corrections, ensuring accurate imaging under various conditions. We appreciate this insightful feedback and will continue to enhance these measures in our future device iterations.

In short, here are our changes to the manuscript:

We have revised the manuscript to incorporate the simulation tests, which can now be found in the supplementary materials.

----- **Manuscript Revision** -----

“For low-density OBS arrays, PS-NUFFT also significantly outperforms DAS in terms of SNR and resolution consistently across different settings (Supplementary Note 16). **Meanwhile, by integrating the pre-migration method [48], PS-NUFFT could compensate for the limited bending-induced image distortion (Supplementary Note 17).”**

----- **Supplementary Note** -----

Supplementary Note 17. Reconstruction performance of PS-NUFFT under bending

conditions.

To validate the reconstruction performance of PS-NUFFT for OBS under bending conditions, three sets of simulations are provided. The simulation setup is the same as supplementary Note 16, except the sensor is under different bending conditions ($R = 100$ mm, $R = 50$ mm and $R = 20$ mm). The image results are shown in Fig. N19. It can be seen that the bending of the OBS will lead to different degrees of distortion of the imaging results (Fig. N19(b), (d) and (f)). The image quality further deteriorates with the decreasing bend radius. The main reason is the acoustic nodes are not at a plane. Fortunately, these distortions could be avoided by incorporating the pre-migration technique [36] into PS-NUFFT. After compensating for the deviation from the positions of acoustic nodes, the image results are restored as shown in Fig. (c), (e) and (g).

Fig. N19 Simulation of reconstructed images before and after compensation. (a) Unbent condition. (b) Image at 100 mm bending radius before compensation. (c) Image at 100mm bending radius after compensation. (d) Image at 50 mm bending radius before compensation. (e) Image at 50 mm bending radius after compensation. (f) Image at 20 mm bending radius before compensation. (g) Image at 20 mm bending radius after compensation.

Q3: How is the device attached to the skin? PVDF membrane is flexible, but not soft enough to conform to the skin. Body motion may introduce the movement of sensor, and air gaps between the sensor and the skin. Did authors observe any motion artefacts during measurement caused by air gaps or other factors?

Response: Thank you for raising queries about how the device attaches to the skin, as well as the potential issues that may arise due to body motion and air gaps. We apply the sensor to the skin using medical double-sided tape which replaces the encapsulation around the optical and acoustic node region (see Fig. R2). While the PVDF membrane is flexible, it isn't sufficiently conformable. We've considered the geometric mismatch between the film and the skin. This is partially improved by introducing a thin and soft PDMS elastomer encapsulation layer that can deform to couple this mismatch, thereby enhancing contact.

For short-term operation, as usual practice, couplant gel can be smeared beneath the probe to further reduce the air gaps. However, traditional ultrasound gel may dry out over time, which could potentially attenuate the signal amplitude. As proposed in reference [2], uncured silicone is a viable alternative for extended use due to its maintained coupling efficiency over time. We understand that vigorous movement can partially introduce air leading to poor ultrasonic coupling or even causing the probe patch to fall off. Recent advancements in the field have introduced a novel bioadhesive hydrogel-elastomer to provide both robust ultrasonic coupling and adhesion, effectively addressing the aforementioned issues [4]. We anticipate such hydrogel-elastomer can also be explored on our probe to eliminate any potential air gaps and maintain excellent long-term performance during vigorous motion.

Fig. R2 The OBS attachment solution

Our current design is intended for smooth skin areas rather than for highly curved regions like human joints. When adhered correctly, we did not observe air-bubble-induced signal loss during measurements, indicating that our probe interface can adjust to subtle skin curvature variations due to translation movement, slight rotational motion, and involuntary motions such as respiration and heartbeat.

Nonetheless, movements causing dynamic skin bending would affect the sensor element's deformation and the positional relationship of blood vessels relative to the skin, thus influencing the optoacoustic signals of blood vessels. This type of motion artifact originates from the dynamic sensor distribution, which we believe can be mitigated by incorporating an additional strain sensor and algorithm, as discussed in response to a previous comment.

In short, here are our changes to the manuscript:

We have enriched our supplementary materials with additional information to address the matter of device attachment and the associated motion artifacts.

----- **Supplementary Note** -----

Supplementary Note 23. Skin bending and motion artifacts

...

Last but not least, a proper adhesion technique can be applied to further minimize the motion

artifacts. The sensor is attached to the skin using medical double-sided tape which replaces the encapsulation around the optical and acoustic node region. While the PVDF membrane is flexible, it isn't sufficiently conformable, hence air gap can present due to the geometric mismatch between the film and the skin. In our design, this is partially improved by introducing a thin soft PDMS encapsulation layer that can deform to couple this mismatch, thereby enhancing contact. For short-term applications, a coupling gel beneath the probe is employed to mitigate the presence of air gaps. However, traditional ultrasound gel may dry out over time, which could potentially attenuate the signal amplitude. As suggested in [52], uncured silicone is a viable alternative for extended use due to its maintained coupling efficiency over time. It is challenging that intense movements can still introduce air gaps, compromise ultrasonic coupling, and potentially cause the sensor patch to detach, all of which can lead to signal loss. Recent advancements in the field have introduced a bioadhesive hydrogel-elastomer [9]. This novel material provides both durable ultrasonic coupling and strong adhesion, effectively addressing the aforementioned issues. Integrating this material into our probe design is expected to effectively eliminate potential air gaps, thereby ensuring robust long-term performance, even under conditions of vigorous motion.

Our current design is intended for smooth skin areas with minimal curvature, making it unsuitable for highly curved regions like human joints. However, when adhered correctly, no air-bubble-induced signal loss was observed during measurements, indicating that the interface can adjust to subtle skin curvature variations due to translation movement, slight rotational motion, and involuntary motions such as respiration and heartbeat.

Nonetheless, movements causing dynamic skin bending would affect the sensor element's deformation and the positional relationship of blood vessels relative to the skin, thus influencing the optoacoustic signals of blood vessels. This type of motion artifact originates from the dynamic sensor distribution, which can be mitigated by incorporating an additional strain sensor and algorithm, as discussed in Supplementary Note 17.

4. In fig. 4h, the color bar shows the blood in the range of 0~1 and tissue in the range of -1~0. What is the criterion? It is not clear why blood signal is higher than 0 but tissue signal is lower than 0 either. How to determine the boundary between the blood and tissue?

Response: In response to the reviewer's comment regarding Fig. 4h, we would like to clarify the criterion used for the color bar, which represents blood and tissue signals. The signal amplitudes of blood are significantly higher than those of the surrounding tissues, thus histogram thresholding [5] was applied to determine the boundary between them. The negative values for tissue signals resulted from subtracting this threshold value from all signal amplitudes, which was done for the purpose of convenient data analysis. However, we acknowledge that employing a color range of -1~0 for tissues may be misleading.

To address this concern, we have updated the color bar in Fig. 4h to display a unified range of 0~1 for both blood and tissue signals, without subtracting the threshold value. This revision should help to avoid potential confusion and provide a clearer representation of the data.

In short, here are our changes to the manuscript:

We have updated the range of the color bar in Fig. 4h to span from 0 to 1.

-----**Manuscript Revision**-----

[1] Gao X, Chen X, Hu H, et al. A photoacoustic patch for three-dimensional imaging of hemoglobin and core temperature[J]. *Nature Communications*, 2022, 13(1): 7757.

[2] Hu H, Ma Y, Gao X, et al. Stretchable ultrasonic arrays for the three-dimensional mapping of the modulus of deep tissue[J]. *Nature Biomedical Engineering*, 2023: 1-14.

[3] Jin, H., Zheng, Z., Liu, S., Zhang, R., Liao, X., Liu, S., & Zheng, Y. (2020). Pre-migration: A general extension for photoacoustic imaging reconstruction. *IEEE Transactions on Computational Imaging*, 6, 1097-1105.

[4] Wang C, Chen X, Wang L, et al. Bioadhesive ultrasound for long-term continuous imaging of diverse organs[J]. *Science*, 2022, 377(6605): 517-523.

[5] Otsu, N., "A Threshold Selection Method from Gray-Level Histograms." *IEEE Transactions on Systems, Man, and Cybernetics*. Vol. 9, No. 1, 1979, pp. 62–66.

Reviewer #2

The authors have addressed most of the concerns that were raised and the work is novel and interesting with sound methodology. I also appreciate the size comparisons to other systems. There is however one discrepancy between the response document and the manuscript regarding the gCNR references. The response document says that the additional helpful references were included, but I could not find them in the manuscript. I read the missing references, and I agree they are helpful as stated in the response document.

Response: We appreciate the reviewer's positive feedback on our revisions and acknowledgement of our work's novelty and methodology. We apologize for the discrepancy and have now rectified this oversight by including the following missing gCNR references in the revised manuscript, which will provide more accurate context for readers.

In short, here are our changes to the manuscript:

In the revised manuscript, the following references pertaining to gCNR are included and can be found as citations [45-47]:

Kempski K M, Graham M T, Gubbi M R, et al. Application of the generalized contrast-to-noise ratio to assess photoacoustic image quality[J]. Biomedical Optics Express, 2020, 11(7): 3684-3698.

Gubbi M R, Gonzalez E A, Bell M A L. Theoretical framework to predict generalized contrast-to-noise ratios of photoacoustic images with applications to computer vision[J]. IEEE Transactions on Ultrasonics, Ferroelectrics, and Frequency Control, 2022, 69(6): 2098-2114.

Gubbi M R, Bell M A L. Theoretical predictions of the generalized contrast-to-noise ratio for photoacoustic images[C]. 2020 IEEE International Ultrasonics Symposium (IUS). IEEE, 2020: 1-4.

REVIEWERS' COMMENTS

Reviewer #1 (Remarks to the Author):

The authors addressed my concerns raised in the last round of review.

For the Reviewer 3's concerns, I think the authors could further improve the manuscript by some minor revisions:

1. As for the first comment, is the terminology 'stethoscope' suitable? I think it is misleading for general readers to call an imaging device 'stethoscope'. Although the authors claimed this device also receives the sound signal like a stethoscope, I suggest that they add more description to explain why they name it 'stethoscope' in the revised manuscript.

2. As for the third comment, Reviewer 3 requires the authors to give a more convincing rationale on the applications needed about the intravascular exogenous agents. I agree with the authors' response to this comment. But why did not the authors add the explanation into the revised manuscript? It would help improve the manuscript quality if this information is added.

Reviewer #2 (Remarks to the Author):

The authors have addressed my concerns.

Reviewer #1

The authors addressed my concerns raised in the last round of review.

For the Reviewer 3's concerns, I think the authors could further improve the manuscript by some minor revisions:

Response: We appreciate your feedback and acknowledgement that we have addressed your concerns from the last round of review. Regarding the issues raised by Reviewer 3, we have made an effort to improve the manuscript based on these comments. Please see our responses to each comment for more detailed information.

1. As for the first comment, is the terminology 'stethoscope' suitable? I think it is misleading for general readers to call an imaging device 'stethoscope'. Although the authors claimed this device also receives the sound signal like a stethoscope, I suggest that they add more description to explain why they name it 'stethoscope' in the revised manuscript.

Response: Thank you for your valuable suggestion. Indeed, our optoacoustic blood 'stethoscope' (OBS) utilizes acoustic signals to garner information about the human body and extracts multiple vital parameters, in a manner like a conventional stethoscope. While we recognize that calling it a 'stethoscope' may understate the device's imaging capabilities, it's important to highlight that the imaging function primarily serves to acquire these vital signs—it is a means of display rather than an end in itself. Labeling our device as a 'stethoscope' aims to provide a more accessible understanding of the information acquisition process and outputs. We sincerely hope this explanation clarifies our choice of terminology. We will add this explanation in our manuscript to avoid any potential misunderstanding.

In short, here are our changes to the manuscript:

We added the following explanation to the introduction:

----- **Manuscript Revision** -----

We have chosen to term our device as a blood 'stethoscope' because, akin to a traditional stethoscope, it 'listens' to the acoustic waves to characterize blood properties. This terminology is used to represent the information acquisition process and its outputs, similar to a conventional stethoscope, while it also signifies the device's unique capability to capture imaging data that serves primarily to acquire these vital signs.

2. As for the third comment, Reviewer 3 requires the authors to give a more convincing rationale on the applications needed about the intravascular exogenous agents. I agree with the authors' response to this comment. But why did not the authors add the explanation into the revised manuscript? It would help improve the manuscript quality if this information is added.

Response: Thank you for your suggestions. We have incorporated the explanation into the revised manuscript as follows:

In short, here are our changes to the manuscript:

We have enriched our supplementary materials by providing a more robust rationale for the need for intravascular exogenous agents, using Photothermal Therapy/Optoacoustic agents as an example.

----- **Supplementary Note**-----

Supplementary Note 3. Bespoke flexible fiber-based solutions for light delivery.

Photothermal Therapy (PTT) presents a promising direction for cancer treatment. This process involves administering photothermal agents (intravascular exogenous agents) that absorb light and convert it into localized hyperthermia through nonradiative transitions, leading to tumor ablation. One significant challenge with PTT is real-time, accurate monitoring of the photothermal agent's concentration and localization in the target tissue. These factors can significantly impact the effectiveness of the treatment and potential side effects. Traditional optical methods such as fluorescence imaging and spectroscopy, commonly used for monitoring agent distribution, face limitations due to shallow tissue penetration and signal interference from surrounding tissues.

To overcome these challenges, we employ optoacoustic imaging in our approach. The intravascular exogenous agents can be designed to absorb light at specific wavelengths that penetrate deeper into tissues. The resulting optoacoustic signals facilitate real-time detection and quantification of the photothermal agent's concentration and distribution, providing vital information for optimizing treatment parameters and minimizing side effects.

Reviewer #2

The authors have addressed my concerns.

Response: We appreciate your time and effort in reviewing our manuscript and are glad to hear that we were able to address your concerns satisfactorily.